# Toxicity Effects and Mechanisms of MgO Nanoparticles on the Oomycete Pathogen *Phytophthora infestans* and Its Host *Solanum tuberosum*

**DOI:** 10.3390/toxics10100553

**Published:** 2022-09-21

**Authors:** Ze-Le Wang, Xi Zhang, Guang-Jin Fan, Yi Que, Feng Xue, Ying-Hong Liu

**Affiliations:** 1College of Plant Protection, Southwest University, Chongqing 400715, China; 2Department of Radiology, Xinqiao Hospital, Army Medical University, Chongqing 400037, China; 3Chongqing Metropolitan College of Science and Technology, Chongqing 402167, China; 4Hanbin Branch of Ankang Tobacco Company, Ankang 725000, China

**Keywords:** magnesium oxide nanoparticles, *Phytophthora infestans*, potato, oxidative stress

## Abstract

Engineered nanoparticles have recently been used for innovation in agricultural disease management. However, both the toxicity effects and mechanisms of nanoparticles in target pathogens and their host plants are still largely unknown. Here, we found that magnesium oxide nanoparticles (MgO NPs) could protect potatoes against *Phytophthora infestans* (*P. infestans*) at a low dosage (50 μg/mL). Through scanning electron microscopy observation, antioxidant enzymes activity measurement, and gene transcriptome analysis, we found that the cell surfaces of *P. infestans* were destroyed, endogenous superoxide dismutase continuously remained in a higher active state, oxidoreductase activity-related gene ontology (GO) terms were enriched with upregulation, and transporter-activity related GO terms and six essential metabolism-related pathways were enriched with downregulation in *P. infestans* after 30 min MgO NPs treatment, whereas only 89 genes were changed without enriched GO and pathways terms, and no change in antioxidant activities and phenylalnine ammonialyase in potato appeared at 6 h post-MgO NPs treatment. Only the “plant hormone signal transduction pathway” was enriched with upregulation under differential expression analysis in potatoes. In conclusion, cell surface distortion, continuous oxidative stress, and inhibitions of membrane transport activity and metabolic pathways were toxic mechanisms of Mg ONPs in *P*. *infestans*, and the “plant hormone signal transduction pathway” was potentially regulated by Mg-ONPs without obviously harmful effects on potato after Mg ONPs exposure.

## 1. Introduction

Potato (*Solanum tuberosum*), which is the world’s third-largest food crop, severely suffers from potato late blight, a devastating disease caused by the oomycete pathogen *Phytophthora infestan* (*P. infestan*) [1]. *Phytophthora infestan*, one of the most important pathogens in agriculture, infects potatoes or tomatoes through the multiplication of sporangia and zoospores and can be transmitted by means of plant survival, rain, soil, and airflow and it is difficult to control [2]. This pathogen belongs to oomycete, which is evolutionarily close to algae. Chemical microbicides are usually used to control potato late blight, but the number of effective microbicides against oomycetes is limited, as the physiology and biochemistry of oomycete are different from those of fungi and most fungicides are ineffective against oomycetes [3]. The long-term use of a finite number of chemical microbicides has also led to the emergence of resistant strains [4] and potential risks to ecosystems and health [5]. On the other hand, genetic resources for effective disease resistance are becoming scarce for the rapid evolution of *Phytophthora* species [6]. Thus, how to deal with the oomycete pathogen is still an unresolved and focal issue in agricultural production.

Recently, nanotechnology is increasingly exploited in a wide range of agricultural applications, including nanofertilizers, nanopesticides, nanomicrobicides, nanosensors, and heavy metal contamination control, to increase food production in a sustainable manner [7,8]. Many different types of inorganic and organic nanomaterials have been found to exhibit excellent antibacterial, antifungal, and antiviral properties on phytopathogenic microbes in the laboratory and even under field conditions [7,8]. To date, many metal oxide nanoparticles (NPs), such as TiO_2_ [9], CuO [10], ZnO [11], AgO [12], and MnO_2_ [13] nanomaterials, have shown toxicity against phytopathogenic pathogens and high potential value for agricultural applications. Generally, the antimicrobial activity of nanomaterials is higher than their bulk form for increasing specific surface area, reducing particle size, and enhancing particle surface reactivity [14]. To date, three toxic mechanisms of metal-based NPs have been found: (1) directly attach or adhere to cell membranes [15], where the membrane can be damaged or initiate internal signaling pathways that damage the cell; (2) release soluble toxic ions, such as Ag+, Cu^2+^, and Zn^2+^ [16]; and (3) produce the reactive oxygen species (ROS) under visible light, which may cause damage to important enzymes or an organism’s genetic material [17]. However, these material characteristics do not completely explain the antimicrobial activity of metal-based NPs yet. Here we take zinc oxide nanoparticles (ZnO NPs), which have been more deeply studied, as an example. First, how the attachment of the cell membrane harms the phytopathogen was largely unknown, as the attachment of ZnO NPs to Hella cells was observed, but lower toxicity was observed [18]. Second, some research also proposed that the concentration of released Zn^2+^ ions from ZnO NPs is not sufficient to generate an antimicrobial effect [19]. Third, although ZnO NPs could generate oxygen radicals or hydrogen peroxide at their surface via oxygen defect sites under visible light [20], previous studies have also shown that ZnO NPs still could sterilize bacteria under darkness [21]. On the other hand, as most previous studies have focused on the material characteristics of metal-based NPs to elucidate their antimicrobial mechanisms, the molecular responses of the pathogen or host plant were largely unknown although these responses were essential to understanding both the biological changes of organisms under metal-based NPs exposure and the toxic mechanisms of NPs on different organisms.

Although nanomaterials provide a novel green way to reduce contamination by chemical agents and drug resistance during plant disease management, the toxicity of inorganic NPs was always an important concern regarding the applications of NPs in the environment [22]. The toxicological aspects of inorganic NPs on the environment or human beings generally include ROS generation and oxidative stress, chemical instability, dissolution of free toxic ions, poisonous chemical composition, left-over chemicals from synthesis, etc. [23]. Thus, a better understanding of the toxicology of inorganic NPs is essential for their application. In fact, crops are one of the most critical components of the agroecological system. Therefore, the perception of the behavior of plants in the presence of nanomaterials also plays an essential role in achieving the goals of sustainable agriculture [24].

Magnesium oxide NPs (MgO NPs), with the advantages of non-toxicity to human cells, ease of availability, and biocompatibility with the environment, function as an antimicrobial agent useful for medical treatment, mosquitocidal action, and textile wastewater treatment [25,26]. Recently, MgO NPs have also been reported as phytopathogenic antagonists against *Ralstonia solanacearum* [27], *Fusarium oxysporum* [28], *Thielaviopsis basicola,* and *Phytophthora parasitica* (*P*. *parasitica*) [15]. The antibacterial activities of the MgO NPs against *Escherichia coli* and *Staphylococcus aureus* have been shown to be affected by the size, pH, concentration, and form of active MgO species [29]. The bactericide activity of MgO NPs further reported that MgO NPs could cause cell membrane leakage, induces oxidative stress, and ultimately leads to bacterial cell death [30]. It was also reported that direct contact interactions between MgO NPs and fungal cells provoked cell morphological changes, a change in zeta potential, and the accumulation of various modes of oxidative stress in MgO NPs-exposed fungal cells [15]. However, different from metal-based NPs of Zn, Ag, or Cu, antimicrobial mechanisms of MgO NPs against phytopathogens are not yet fully understood, especially against oomycete. On the other hand, MgO NPs were also used as a nanofertilizer to enhance the growth of maize [31] and tobacco [32]. However, the potentially toxic effects of MgO NPs on the plant are also still unclear.

In this study, we assessed the direct toxic activity of MgO NPs on *P. infestans* and the protection ability of potatoes against *P. infestans*. Additionally, the destruction of *P. infestans* hyphae using MgO NPs was confirmed through a scanning electron microscope (SEM). The transcriptome sequencing was used to systematically uncover the potential toxicity effects and mechanisms of MgO NPs on *P. infestans* and potatoes. Finally, the enzyme activity analysis of superoxide dismutase (SOD), catalase (CAT), peroxidase (POD), and phenylalanine ammonialyase (PAL) was used to evaluate the effects of MgO NPs on plant physiology and biochemistry. SOD was also detected in *P*. *infestans* to evaluate the oxidative stress of MgO NPs. Ultimately, we proposed that the continuous ROS stress, disruption of the cell surface, and inhibition of transport capacity and essential metabolic pathways were the main toxicity mechanisms for toxicity against *P. infestans*, and MgO NPs had no significant harmful effect on potatoes, but could potentially promote plant growth and disease resistance through modulating the plant hormone signal transduction pathways. Our study not only provides a potential oomycete disease management strategy in agricultural applications, but also extends the understanding of the novel toxic effects and mechanisms of metal NPs on the pathogen and its host plant.

## 2. Materials and Methods

### 2.1. Characterization of MgO NPs

The MgO NPs were synthesized according to the previous report [33]. Briefly, aqueous solutions of Mg(NO_3_)_2_ and ammonia solution were mixed at room temperature, and a precipitation reaction occurred between the Mg^2+^ and OH^−^ ions during stirring for 12 h, resulting in the formation of magnesium hydroxide. Furthermore, the resulting hydroxide precipitate was calcined at 500 °C, which led to the decomposition of Mg(OH)_2_ to MgO NPs. MgO NPs were dispersed in ultrapure water and spread by using an ultrasonic bath sonicator before SEM, transmission electron microscopy (TEM), and zeta potential analysis. The morphology and agglomeration of MgO NPs were visualized by SEM with ZEISS Sigma 300 (Germany) after the samples were conductively coated by gold sputtering (<10 nm). The average size of MgO NPs was calculated using the TEM for approximately 100 NPs. The crystalline morphology and structure of MgO NPs were determined using TEM/HRTEM performed on an FEI Tecnai G2 F20 [34] Scanning Transmission Electron Microscope with an accelerating voltage of 300 kV. XRD was conducted on an X-ray diffractometer (D8 Advance of Bruker, Germany) with Cu Kα radiation (λ = 0.1541844 nm, 40 kV, and 40 mA) to determine the crystallographic structure and crystalline size of the MgO NPs. The zeta potentials of MgO NPs were measured by using a Malvern Zetasizer Nano ZS90 (UK) at pH = 7.

### 2.2. Phytophthora infestans and Potato Culture 

*Phytophthora infestans* strain “88069” was routinely grown at 18 °C in the dark on rye sucrose agar (RSA) [35] medium in this study [36]. The potato cultivar “FAVORITA” was routinely cultured at 22 °C in a plant growth chamber with 16 h of light (8000–10,000 lux) and 8 h of dark daily.

### 2.3. Antimicrobial Activity Assay of MgO NPs to P. infestans

To evaluate the in vitro antimicrobial activity of MgO NPs, different concentrations of MgSO_4_, bulk MgO and MgO NPs were added to the RSA culture, which was poured into Petri dishes. Additionally, a 6 mm diameter of hyphae block was added to Petri dishes for culture. The colony diameter of *P. infestans* was measured when the control (CK) Petri dishes were nearly full of *P. infestans* and the inhibition rate was calculated as flow to evaluate the antimicrobial activity for each treatment:Inhibition rate (%)=(R−r)/R × 100%
where R is the average diameter of the control group (CK), and r is the average diameter of the treatment group.

To evaluate the in vivo protection activity of MgO NPs against *P. infestans* in vivo, detached potato leaves were sprayed with MgO NPs suspension or sterile water. Additionally, different treatment leaves were inoculated with fresh *P. infestans* hyphae cubes of the same size. After seven days of inoculation, the disease lesions were measured and compared to evaluate the attenuation of potato late blight by MgO NPs.

### 2.4. Obervation of Morphological Changes in P. infestans Using SEM

To further explore the morphological changes in the *P. infestans* exposed to MgO NPs, the morphologies of the *P. infestans* were investigated using SEM. The fresh *P. infestans* hyphae cubes were cultured on the RSA medium, which was coated with sterilized cellophane. After mycelia covered most of the area on the Petri dishes, *P. infestans* were sprayed with MgO NPs (50 mg/L) or sterile water. After 6 h, mycelia of *P*. *infestans* were collected, gently washed with PBS solution (pH 7.4), and incubated with 2.5% glutaraldehyde for 4 h. Subsequently, the hyphal samples were then dehydrated by using a series of concentrations (30–100%) of ethanol. After being air-dried naturally, the samples were conductively coated by gold sputter (<10 nm) and subjected to SEM (FEI Quanta 200, Eindhoven, The Netherlands) operated at accelerating voltages at 30 kV. 

### 2.5. SOD, POD, CAT, and PAL Activity Assay

The activity of PAL was used as an important metabolic process indicator in potatoes. The total SOD activity in potato leaves was determined using a kit (A001-1, Nanjing Jiancheng Bioengineering Institute, Nanjing, China) based on the nitrobluetetrazolium method according to the manufacturer’s instruction, as shown in a previous report [37]. The POD activity was measured by using a POD assay kit (A084-3-1; Nanjing Jiancheng Bioengineering Institute, Najing, China) on the basis of guaiacol oxidation at 470 nm by H_2_O_2_ according to the manufacturer’s instruction, as shown in a previous report [38]. Similarly, the activity of CAT was also measured using a CAT assay kit (A007-1; Nanjing Jiancheng Bioengineering Institute, China) as seen in a report [38]. The PAL can catalyze L-phenylalanine to produce trans-cinnamic acid and ammonia, and the maximum absorption peak of trans-cinnamic acid is 290 nm. PAL activity was calculated by measuring the increase in the OD value to 290 nm using a PAL assay kit (E-BC-K522-S; Elabscience Biotechnology Co. Ltd, China). As there was no POD, CAT, and PAL activity in *P. infestans* through test kits, the SOD activity of *P. infestans* was used as the antioxidant system indicator in *P. infestans*. 

### 2.6. RNA Extraction and Sequencing

The fresh *P. infestans* hyphae cubes were cultured on the RSA medium, which was coated with sterilized cellophane. After mycelia covered most of the area on the Petri dishes, *P. infestans* was treated with MgO NPs (50 mg/L) or sterile water. For the sample of *P. infestans*, the fresh *P. infestans* hyphae were sampled after 30 min. Six-week-old potatoes were treated with MgO NPs (50 mg/L) or sterile water in daylight. The samples of potato leaves were detached and collected 6 h after treatment. All samples were quick-frozen with liquid nitrogen. The RNA extraction, RNA quantity assessment, cDNA library construction, and paired-end RNA-seq were all operated by Majorbio Bio-pharm Biotechnology Co., Ltd. (Shanghai, China) as previously described [39]. Briefly, total RNA was isolated with TRIzol (Invitrogen, USA); the RNA quantities were assessed using agarose gel electrophoresis, a NanoDrop 2000 spectrophotometer (Thermo Scientific, Wilmington, United States), and an Agilent 2100 bioanalyzer (Agilent Technologies, Santa Clara, United States). Only high-quality RNA samples (OD 260/280 = 1.8~2.0, OD 260/230 ≥ 2.0, RIN ≥ 6.5, 28S:18S ≥ 1.0, RNA concentrations ≥100 ng/μL, total amount ≥ 2 μg) were used to construct the sequencing library. The cDNA library was constructed using extracted mRNA according to a TruseqTM RNA sample prep kit (Illumina, SAN DIEGO, USA) and sequenced using the Illumina NovaSeq 6000 System. The processing of original images to sequences, base-calling, and quality value calculations were performed using the Illumina GA Pipeline, in which 150bp paired-end reads were obtained. A Perl program was written to select clean reads by removing low-quality sequences, reads with more than 5% N bases (unknown bases), and reads containing adaptor sequences.

### 2.7. Transcriptome Data Analysis

The raw paired-end reads were trimmed and quality controlled using fastp (https://github.com/OpenGene/fastp, version 0.19.5, accessed on 18 July 2022) with default parameters. Then, clean reads were separately aligned to the reference genome in orientation mode using HISAT2 (version 2.1.0) software [40]. The mapped reads of each sample were assembled using StringTie [41] in a reference-based approach.

To identify significantly differentially expressed genes (DEGs) between two different samples/groups, the expression level of each gene was calculated according to the transcripts per million reads (TPM) method. RSEM [42] (version 1.3.3) was used to quantify gene abundances. Essentially, differential expression analysis was performed using the DESeq2 (version 1.24.0). Genes with |log_2_(fold change)| > 1 and adjusted *p*-value < 0.05 were considered to be strictly significant DEGs. Genes with |log_2_(fold change)| > 1 and *p*-value < 0.05 were considered to be loosely significant DEGs.

Clusters of Orthologous Groups of proteins (COG) categories for DEGs were assigned using the STRING database (https://string-db.org/; version 11.5, accessed on 18 July 2022). Functional-enrichment analysis including GO (http://www.geneontology.org; version 2020.0628, accessed on 18 July 2022) and Encyclopedia of Genes and Genomes (KEGG) (http://www.genome.jp/kegg/; version 2020.07, accessed on 18 July 2022), was performed to identify which DEGs were significantly enriched in GO terms and metabolic pathways at adjusted *p*-value ≤ 0.05 compared to the whole-transcriptome background, respectively. GO functional enrichment and KEGG pathway analysis were carried out using Goatools (version 0.6.5) [43] and KOBAS (version 2.1.1) [44], respectively.

### 2.8. RNA Isolation, cDNA Synthesis, and Quantitative RT-PCR

Total RNA was isolated using TRIzol (Invitrogen, Carlsbad, CA, USA) according to the manufacturer’s instructions. The cDNA was obtained from the total RNA by using the PrimeScript RT Reagent Kit (RR037A, TaKaRa, Otsu Shiga, Japan) according to the manufacturer’s instructions. RNA quantitative real-time PCR (qRT-PCR) was performed with gene-specific primers (Appendix A) using the SYBR Green Color qPCR SuperMix Kit (E090-01A, NovoStart, Nanjing, China).

## 3. Results

### 3.1. Characterization of MgO NPs

The antimicrobial activities of NPs on microorganisms are frequently associated with their chemical and physical characteristics. Thus, the morphological characteristics and size of MgO NPs were firstly investigated in this study. SEM data (Figure 1A) showed that MgO NPs were irregularly spherical and coupled with some sliced shells. TEM data (Figure 1 B), which also verifies the SEM data, showed that the average size of MgO NPs was approximately 341 nm and the NPs also have irregularly spherical with some sliced shells. The lattice fringes of MgO NPs were also visualized using HRTEM (Figure 1C). The Zeta potential of MgO NPs typically ranged from +14.9–+17 mV. The XRD pattern (Figure 1D) showed that several sharp peaks, which were located at 2θ of 36.95°, 42.92°, 62.30°, 74.76°, and 78.61°, were assigned to the (111), (200), (220), (311), and (222) crystallographic planes of the face-centered cubic (FCC)-structured MgO Nano powders [Joint Committee on Powder Diffraction Standards (JCPDS) file no. 89-7746]. No other peaks were detected in the XRD spectrum, indicating the high purity of the obtained MgO NPs, in accordance with previous reports [15,45,46]. The crystalline sizes of NPs are 85.6 nm, 77.9 nm, 62.5 nm, 65 nm, and 69.6 nm, respectively.

### 3.2. The Antimicrobial Activity of MgO NPs against Phytophthora infestans

The antimicrobial activity of MgO NPs on *P. infestans* was firstly evaluated by measuring the vegetative growth of *P. infestans* in the culture dish under different concentrations of MgSO_4_, bulk MgO and MgO NP suspension treatments. In this process, MgO NPs have the same number of moles of magnesium elements with bulk MgO or MgSO_4_ for each concentration. As shown in Figure 2B, with an increase in bulk MgO and MgO NPs concentrations, the inhibitions of bulk MgO and MgO NPs against *P. infestans* were also increased, and the inhibition rate of MgO NPs was much higher than bulk MgO at each of the same concentration. However, there were no obvious inhibition increases in MgSO_4_ treatments with the increase in the magnesium ion concentration. To determine whether the MgO NPs could protect potatoes against *P. infestans*, the curative activities of MgO NPs towards potato late blight were also evaluated on the potato. As shown in Figure 2C and2D, after spraying MgO NPs on the potato leaves, the lesion size of leaf disease was prominently reduced in comparison with the water (CK) groups.

To characterize the toxic effects of the MgO NPs attachment on the hyphae in *P. infestans*, the morphological changes of *P. infestans* cells by MgO NPs were further monitored through biological SEM. The presence of MgO NPs on the hyphae was visualized through energy-dispersive spectrometry. As shown in Figure 2E, the sterile water-treated mycelia maintained a full, uniform, and well-developed tube-like hyphae structure without magnesium element (Figure 2F). On the contrary, treatments with MgO NPs on *P. infestans* led to obviously unfavorable changes and collapsed morphologies (Figure 2G). Precisely, the magnesium element was also monitored through energy-dispersive spectrometry (Figure 2H) on the hyphae surface. All these results showed that MgO NPs could also adhere to the surface of *P. infestans* and result in the destruction of the cell surface of *P. infestans*. 

### 3.3. Overview of Transcriptome Profiles of P. infestans after MgO NPs Treatment at Early Stage

To roundly investigate the toxic effects of MgO NPs on the *P. infestans*, *P. infestans* were collected and subjected to transcriptome analysis after treatment with water or MgO NPs at 30 min into the mycelia growth stage. Three MgO NP-treated and three water-treated samples of *P. infestans* were subjected to Illumina sequencing and a total of 287,078,508 raw reads were obtained from the six libraries (Appendix A). For further analysis, unknown (the proportion of undetermined bases >10%), adaptor-containing, and low-quality reads were filtered out, and a total of 284,076,856 clean reads were obtained. The average GC content was 55.7–55.86%, the genome mapped ratio was 94.04–94.43% and Q30 values were 94.84–95.41% in these samples (Appendix A). The saturation curves (Appendix A), coverage, and expression distributions (Appendix A) of all the transcriptome data in six libraries met the sequencing quality requirements.

A total of 18,963 high-quality unigenes of *P. infestans* in six samples were mapped and annotated through blasting with NR, Swiss-Prot, Pfam, COG, and GO databases (Appendix A). The Venn diagram (Figure 3A) showed that there were 13,212 overlapping genes; the control-only expressed 230 genes and MgO NPs-treatment-only expressed 217 genes. The criteria of the adjusted *p*-value < 0.05 and fold changes <0.5 or >2.0 were used as the cutoff of the significant DEGs. Compared with the control, 255 DEGs were found, with 60 differentially up-regulated genes (DUGs) and 195 differentially down-regulated genes (DDGs) (Figure 3B and Appendix A). The principal component analysis (PCA) showed that there were significant differences in gene expression between MgO NPs and water treatments (Appendix A). All 7 representative genes from DEGs had similar fold changes between RNA-Seq analyses and qRT-PCR (Appendix A), suggesting that our RNA-Seq data were reliable.

### 3.4. Transcriptome Change Analysis of P. infestans after MgO NP Exposure at Early Stage

To further understand the gene function and evolution of DEGs, COG was used to predict and classify the DEGs (Appendix A). Based on the sequence homology, these DEGs of *P. infestans* were finally mapped into 21 different COG categories (Appendix A). With the exception of the “function unknown” (114 unigenes) category, other categories of DEGs supported the hypothesis MgO NPs that could change the “cellular processes and signaling” (41 unigenes), “information storage and processing” (17 unigenes) and “metabolism” (57 unigenes) of *P. infestans*. In particular, most “metabolism” genes (39/57) were downregulated, implying that the metabolism of *P. infestans* was suppressed by MgO NPs.

To understand the functions of DEGs in *P. infestans* under MgO NP exposure at an early stage, the GO, which provides an ontology of defined terms representing gene properties, was annotated for all DEGs (Appendix A). All DEGs were functionally characterized into GO categories (Appendix A) and a total of 147 genes related to the “biological process”, 155 genes related to the “cellular component” and 251 genes related to “molecular function” (Appendix A). To facilitate the analysis of gene function changes, DUGs and DDGs were separately subjected to enrichment analysis, as shown in a previous report [39]. The GO enrichment analysis of DUGs (Figure 4A) suggests that the significant enrichment term with the largest number of genes was “oxidoreductase activity” (GO:0016491). Other significant enrichment terms of DUGs included “solute:sodium symporter activity” (GO:0015370), “solute:cation symporter activity” (GO:0015294), “sodium:phosphate symporter activity” (GO:0005436), “sodium ion transmembrane transporter activity” (GO:0015081), “oxidoreductase activity, acting on peroxide as acceptor” (GO:0016684), “antioxidant activity” (GO: GO:0016209), “oxidoreductase activity” (GO:0016491), “peroxidase activity” (GO:0004601), “symporter activity” (GO:0015293), “active ion transmembrane transporter activity” (GO:0022853), “response to oxidative stress” (GO:0006979) and “monovalent inorganic cation transmembrane transporter activity” (GO:0015077). All significantly enriched GO terms for DUGs were associated with the activity to deal with oxidative stress and the ion transmembrane transport (Figure 4A), which suggests that MgO NPs could generate oxidative stress and an imbalance of ions on *P*. *infestans*. The GO enrichment analysis of DDGs (Figure 4B) showed that the significant enriched GO terms with the largest ratio of the population and number of genes were “transmembrane transporter activity” (GO:0022857) and “transporter activity” (GO:0005215). Other significantly enriched GO terms of DDGs included the “branched-chain amino acid biosynthetic process” (GO:0009082), “pyridoxal phosphate binding” (GO:0030170), “vitamin B6 binding” (GO:0070279), “isoleucine biosynthetic process” (GO:0009097), “isoleucine metabolic process” (GO:0006549), “vitamin binding” (GO:0019842), “branched-chain amino acid metabolic process” (GO:0009081), “cellular amino acid biosynthetic process” (GO:0008652) and “alpha-amino acid metabolic process” (GO:1901605). These enriched GO terms in DDGs, which perform a function in transporter activity, suggested that the transporter activity of *P*. *infestans* was inhibited by MgO NPs, which was also inconsistent with the distortion of the *P*. *infestans* cell surface (Figure 2E,F). On the other hand, these enriched GO terms, which perform a function in synthesis and metabolism, re-implied that several synthesis and metabolism functions of *P*. *infestans* are also inhibited by MgO NPs.

To further understand specific metabolite synthesis processes, relevant gene functioning, and multiple gene interactions at the transcriptome level, the KEGG pathways were also annotated (Appendix A) and clustered (Appendix A) for DEGs. No significant enrichment KEGG terms were found in DUGs. “Glycine, serine and threonine metabolism” (map00260), “Alanine, aspartate, and glutamate metabolism” (map00250), “Tyrosine metabolism” (map00350), “Butanoate metabolism” (map00650), and “Valine, leucine, and isoleucine biosynthesis” (map00290) were significantly enriched for DDGs (Figure 4C). This result provides further solid evidence that MgO NPs could stunt the metabolism of *P. infestans*. No significant enrichment KEGG pathways were found in the genes, which were only detected in MgO NP-treated samples. However, the KEGG pathways (Figure 4D) of “Pentose and glucuronate interconversions” (map00040), “Glycerolipid metabolism” (map00561), and “Glycolysis/Gluconeogenesis” (map00010) were also significantly enriched in genes, which were only expressed in CK. This result additionally supported that the energy-related metabolism of *P. infestans* was also stunted by MgO NPs. 

### 3.5. Overview of Transcriptome Profiles of Potato after MgO NPs Treatment

To investigate the effect of MgO NPs on the plant host of *P. infestans*, 6-week-old potatoes were sprayed with MgO NPs (50 mg/L) or sterile water (control). The leaves of potatoes were collected and subjected to transcriptome sequencing after being treated with water or MgO NPs after 6 h. In this study, three MgO NP-treated and three water-treated samples were subjected to Illumina sequencing, and a total of 285,361,050 raw reads were obtained from the six libraries. For further analysis, a total of 282,860,184 clean reads were obtained as described in the Materials and Methods section (Appendix A). Saturation curves (Appendix A), coverage and expression distributions (Appendix A) of all transcriptome data also met the sequencing quality requirements, as described previously.

A total of 32,917 high-quality unigenes of potatoes were mapped and annotated (Appendix A). The Venn diagram (Appendix A) showed that there were 19,334 overlapping genes, the control-only expressed 510 genes, and MgO NP- -only expressed 788 genes. The PCA analysis showed that there were no significant differences between MgO NPs and water treatments (Appendix A). In fact, there were only 89 DEGs between the MgO NPs-treatment and control, when the cutoff criterion was the same as the transcriptome analysis for *P. infestans* (Figure 5A, Appendix A). To further understand the potential effects of MgO NPs on potatoes, a loosed cutoff criterion, which is a *p*-value < 0.05 and fold changes of <0.5 or >2.0, was chosen and resulted in 695 DEGs (Figure 5B, Appendix A), including 237 DUGs and 458 DDGs. All 7 representative genes from DEGs also had similar fold changes between qRT-PCR (Appendix A) and RNA-Seq analyses, suggesting that our RNA-Seq data for potatoes were also reliable.

### 3.6. Transcriptome Change Analysis of Potato after MgO NPs Treatment

It was found that no significant GO terms or KEGG pathways were significantly enriched for 89 DEGs, which were calculated using a strict threshold. The GO term enrichment analysis for genes that were only expressed in MgO NP treatments, showed that “microtubule” (GO:0005874), “supramolecular fiber” (GO:0099512), “polymeric cytoskeletal fiber” (GO:0099513), “supramolecular polymer” (GO:0099081) and “microtubule binding” (GO:0008017) were significantly enriched GO terms (Figure 6C). There was no enrichment GO term for genes, which were only expressed in CK. 

To uncover the potential modulation of MgO NPs on potatoes, 695 DEGs with loose thresholds were also annotated using COG, GO, and KEGG (Appendix A) as previously described. These DEGs of potatoes were finally mapped on 20 different COG categories (Appendix A). With the exception of the S categories of COG, which are “function unknown” (342 unigenes), other categories of DEGs supported that MgO NPs could change the “cellular processes and signaling” (151 unigenes), “information storage and processing” (58 unigenes) and “metabolism” (140 unigenes) of potato.

Additionally, GO enrichment analysis was performed for DUGs and DDGs of potatoes, respectively. The GO-enriched terms of potato DUGs included the “protein disulfide oxidoreductase activity” (GO:0015035), “response to auxin” (GO:0009733), “oxidoreductase activity, acting on a sulfur group of donors” (GO:0016667), “cell redox homeostasis” (GO:0045454), “regulation of biological quality” (GO:0065008), “disulfide oxidoreductase activity” (GO:0015036), “obsolete cell” (GO:0005623), “response to hormone” (GO:0009725), “response to endogenous stimulus” (GO:0009719), and “regulation of hormone levels” (GO:0010817) (Figure 6A). Correspondingly, 34 significantly enriched GO terms were enriched for potato DDGs, as shown in Appendix A. Among these 34 significantly enriched GO terms, 24 GO terms belonged to the biological process (BP), 9 GO terms were for the molecular function (MF) and only one GO term belonged to the cellular component (CC). The top 20 enriched GO terms are shown in Figure 6B. The genes, which were only detected in MgO NPs treatment or sterile water samples, were also functionally characterized into GO categories. Together, these GO enrichment analyses suggested that MgO NPs also potentially caused oxidative stress on potatoes, but there was no obvious disruption of the plant membrane transporter activity or metabolism suppression as with *P. infestans*.

Finally, KEGG enrichment analyses were performed with loose DDG and DUG thresholds and showed that only plant hormone signal transduction (map04075) was significantly enriched in DUGs. Figure 6D shows that multiple genes, which were responsible for plant growth signaling transduction (auxin, cytokinin, gibberellin, and abscisic acid-mediated pathways), were significantly up-regulated. The up-regulated hormone signal pathway is mainly associated with the vegetative growth of the plant. On the other hand, there were no significantly enriched KEGG pathways in genes, which were only detected in the control and MgO NP treatments. Thus, our transcriptome analysis of potatoes supported the hypothesis that there were no obvious harmful effects of MgO NPs (50 mg/mL) on potatoes. In fact, there were no obvious physiological changes after spraying MgO NPs (250 mg/mL) for 3 days (Appendix A). 

### 3.7. Evaluation of the Antioxidant and Important Metabolism Enzyme Activities in P. infestans and Potato under MgO NPs Exposure

As there was no POD, CAT, or PAL activity observed in *P*. *infestans* using test kits, the SOD activity of *P*. *infestans* was used as the antioxidant system indicator in *P*. *infestans*. To further reconfirm the ROS stress of *P. infestans* by MgO NPs, the SOD activities of *P. infestans* were detected after 1, 3, and 6 h of MgO NP (50 mg/L) treatment. As shown in Figure 7A, the SOD activities of MgO NP treatment samples were continuously higher than those of control samples. Thus, we proposed that MgO NPs could continuously induce oxidative stress to the oomycete pathogen and disturbed the oxidant-antioxidant homeostasis in *P. infestans*. To obtain the changes in potato biochemical indices under MgO NPs treatment, six-week-old potatoes were sprayed with MgO NP (50 mg/L) or sterile water, and the leaves were sampled after 6 h. The activity of SOD, POD, and CAT was used as the antioxidant system indicators. PAL catalyzes the first step of the phenylpropanoid pathway, which produces precursors to a variety of important secondary metabolites in plants. As shown in Figure 7B, there were no obvious differences between the MgO NPs treatment and control groups for SOD, CAT, and POD activities. Although a small number of oxidoreductase activity-related genes were up-regulated 6 h after spraying with MgO NPs, this result supported the hypothesis that the potato could maintain redox homeostasis under MgO NPs stress, which was also in accordance with the results of the transcriptome analysis on potato, showing no harmful effects. Thus, we speculated that the oxidative stress from Mg ONPs was not enough to toxify plants. At the same time, Figure 7C shows that PAL, which is essential for plant secondary metabolism, was also not affected by MgO NPs, corresponding with the finding that the metabolism pathways were not inhibited in the transcriptome analysis on potatoes. 

## 4. Discussion

At present, researchers often face the problem of striking a balance between the positive therapeutic effect of NPs and the side effects related to their toxicity [47]. Thus, the toxic effects and mechanisms of the different NPs on phytopathogens and their host plants are especially crucial during agricultural and environmental systems applications. In this study, we demonstrated that the MgO NPs (50 mg/L) could both directly inhibit *P. infestans* and attenuate potato late blight (Figure 2) and had no obvious toxic effects on potatoes (Figure 6, Figure 7 and Appendix A). The toxicity mechanism of MgO NPs in *P. infestans* was explained through the observation of microscopic changes on the cell surface (Figure 2E–G), antioxidant activity detection (Figure 7), and transcriptomic analysis (Figure 4). Through GO and KEGG analysis of the DUGs and DDGs of *P*. *infestans*, we found that many oxidoreductases and several ion transport-related GO terms were up-regulated and enriched, and many transport GO terms and metabolism-related pathways were down-regulated and enriched (Figure 4). Thus, we deemed that continuous oxidative stress, cell surface disruption, and the inhibition of membrane transport capacity and essential metabolic pathways were key toxicity mechanisms of MgO NPs in *P. infestans*, while the antioxidant enzyme activity, PAL activity, and transcriptomic changes in potato under MgO NPs treatment supported that the stress from MgO NPs was not sufficient to disturb the redox homeostasis (Figure 7) and essential metabolic pathways of potato but could potentially modulate the plant hormone signal transduction pathways (Figure 6), which could promote plant disease resistance and plant growth. Thus, we also speculated that the strong maintenance ability of redox homeostasis and different cell surface structure of plants provided protection against toxicity from nanomaterials. In summary, we verified the protective activity of MgO NPs (50 mg/L) in potatoes against *P*. *infestans*, completely displayed molecular and biochemical changes in *P*. *infestans* and potatoes due to MgO NP exposure, uncovered the toxic mechanisms of MgO NPs in *P*. *infestans*, and found the potential novel benefits of spraying MgO NPs onto potatoes. This study will both extend our understanding of NPs potentially being adopted as an effective strategy for preventing phytopathogen infections and augment the toxic mechanisms and effects of NPs on different organisms in agricultural fields.

### 4.1. The Synthesis and Characterization of MgO NPs

The MgO NPwereas synthesized, as previously reported [33], using a green approach, in which the catalyst could be reused in subsequent reactions with consistent activity. The morphological characteristics of our MgO NPs were irregularly spherical with some sliced shells (Figure 1A,B), which was also in accordance with a previous report [33]. However, this is different from the MgO NPs, which were used in the inhibition of *P*. *parasitica* [15]. This sharply sliced shell structure may lead to great antimicrobial activities. On the other hand, the high purity of the obtained MgO NPs was similar to many other synthetic MgO NPs described in previous studies in the XRD pattern [15,45,46].

### 4.2. The Antimicrobial Activities and Mechanisms of MgO NPs on P. infestans

With the exception of inhibiting bacteria [48,49] and fungi [28], MgO NPs were first found to have antimicrobial activities against *P*. *infestans* (Figure 2), which is largely different from bacteria and fungi [3]. Although recent research has reported that MgO NPs could inhibit *P*. *parasitica*, which is an important oomycete pathogen in tobacco, the inhibition rate of MgO NPs to *P. infestans* could nearly achieve 100% under the concentration of 50 mg/L (Figure 2), which was much lower than the concentration (500 mg/L) of the complete inhibition of *P*. *parasitica* by MgO NPs [15]. This means that our synthetic MgO NPs had higher toxicity than previous MgO NPs or MgO NPs were more applicable to the inhibition of *P*. *infestans* than *P*. *parasitica*. The inhibition rate of MgO NPs was much higher than that of bulk MgO at each of the same concentrations in our study, which was similar to a previous experimental result that MgO NPs have higher toxicity than bulk MgO against *P*. *parasitica* [15]. Interestingly, MgSO_4_ treatments showed nearly non-toxic activity in our study against *P*. *infestans*, indicating that the dissolved magnesium ion from MgO NPs cannot be used as a toxic mechanism for MgO NPs.

Transcriptome analysis was used to understand both the biological processes of *P*. *infestans* under MgO NP exposure and the antimicrobial mechanisms of MgO NPs. We found a total of 255 DEGs, with only 60 up-regulated genes and as many as 195 down-regulated genes (Figure 3B, Appendix A) at 30 min post-MgO NPs spraying. Oxidative stress and ion transmembrane transport-related GO terms were significantly enriched in up-regulated transcripts. This result illustrated that MgO NPs could generate oxidative stress, which identified reactive oxygen species production in *P*. *parasitica* under MgO NP exposure [15]. In fact, a previous report [50] also showed that MgO NPs had potent photocatalytic activity under UV radiation and sunlight with the generation of reactive oxygen species (ROS), which could produce oxidative stress on *P. infestans*. The ion transmembrane transport-related genes may be triggered by the dissolved Mg^2+^ from MgO NPs as a large number of magnesium ions around the surface of the metal oxide nanomaterials have been detected in previous studies [19]. On the other hand, enriched GO terms in down-regulated genes supported that the transporter activity of *P*. *infestans* was inhibited by MgO NPs, which result from oxidative stress or cell surface distortion. Through KEGG enrichment analysis, as many as ten metabolism pathways were stunted by MgO NPs (Figure 4C,D), which is a novel discovery in the toxic mechanisms of metal-based NPs. Combined with the result of continuously higher SOD activities in *P*. *infestans* under MgO NP treatment, we proposed that continuous oxidative stress, cell surface distortion, and the inhibition of membrane transport capacity and essential metabolic pathways were toxic mechanisms of MgO NPs on *P. infestans*.

### 4.3. The Effects of MgO NPs on Potato

Transcriptome analysis was also used to systematically explore the effects of MgO NPs on the host plant of *P*. *infestans*: potato. However, there were only 89 DEGs (Figure 5A) in MgO NPs-treated potatoes with no significantly enriched GO terms or KEGG pathways in all DEGs, DUGs, and DDGs. To uncover the potential molecular effects of MgO NPs on potatoes, a loos cutoff criterion was used to enlarge the number of DEGs and resulted in 695 DEGs for further analysis (Figure 5B). We also found that oxidoreductase activity-related GO terms were enriched by up-regulation in loose DEGs, which suggested that MgO NPs could also generate ROS stress on potato leaves. However, the evaluation of the antioxidant and important metabolism enzyme activities in potatoes (Figure 7B,C) suggested that the ROS from the MgO NP were not sufficient to disrupt redox homeostasis in potatoes, which is different from the result of *P*. *infestans*. Although ROS could generally induce programmed cell death [51] under biological attack, an appropriate amount of ROS also plays a crucial role as a signaling molecule that promotes plant growth and triggers subsequent downstream reactions when the plants are healthy [52]. Interestingly, plant hormone signal transduction pathways (Figure 6), which could promote plant growth and plant disease resistance, were enriched by upregulation in our transcriptome analysis. Specifically, multiple genes that were responsible for plant growth signaling transduction (auxin, cytokinin, gibberellin, and abscisic acid-mediated pathways) were significantly up-regulated (Figure 6D), which addressed and might explain a previous report that MgO NPs could promote tobacco growth [32]. This result could also be supported by the finding that the MgO NPs-treatment-only genes were enriched in the “microtubule” (GO:0005874), “supramolecular fiber” (GO:0099512), “polymeric cytoskeletal fiber” (GO:0099513), “supramolecular polymer” (GO:0099081) and “microtubule binding” (GO:0008017), which suggested plant growth. On the other hand, the up-regulation of NPR1 in our result might explain the previous report that the MgO NPs could induce systemic resistance in tomatoes against bacterial wilt disease [53]. Except for fertilizer supplementation, here, we found another novel possible way to promote plant growth through MgO NP exposure. All of these results also supported that MgO NPs (50 mg/L) exposure has no obvious harmful effects on potatoes.

## 5. Conclusions

Little is known about the nanoparticle–oomycete phytopathogen and nanoparticle–host–plant interactions when exposed to agricultural and environmental systems. Here, we demonstrated that the MgO NPs could directly inhibit *P*. *infestans* and attenuated potato late blight. The continuous ROS stress, cell surface disruption, and inhibition of both membrane transport capacity and essential metabolic pathways in *P. infestans* were three key antimicrobial mechanisms of MgO NPs against *P. infestans*. The stress from MgO NPs could not disturb the redox homeostasis and general metabolic pathways in potatoes, but could potentially modulate plant hormone signal transduction pathways to promote plant growth and disease resistance.

## Figures and Tables

**Figure 1 toxics-10-00553-f001:**
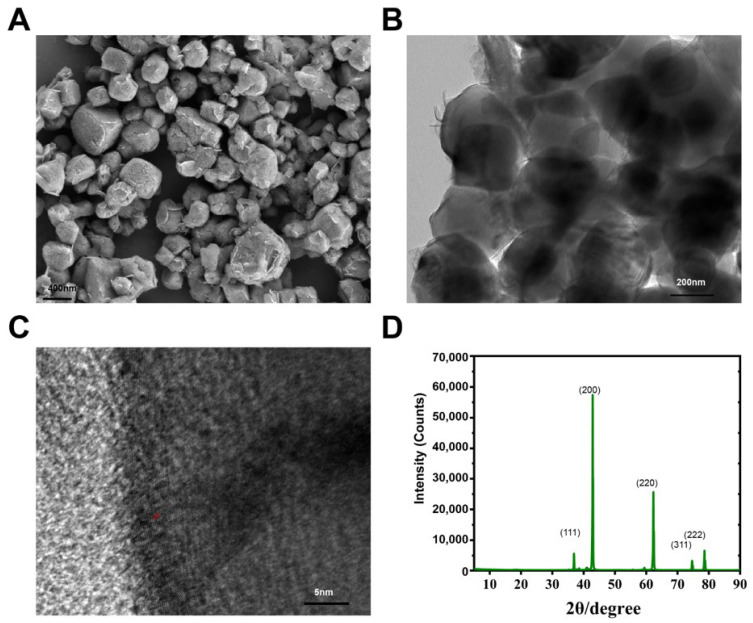
Morphological characteristics and size of MgO NPs. (**A**) Representative scanning electron microscope of prepared MgO NPs. (**B**) Representative transmission electron microscopy of prepared MgO NPs. (**C**) High-magnification view of MgO NPs. (**D**) X-ray diffraction (XRD) survey spectrum of MgO NPs.

**Figure 2 toxics-10-00553-f002:**
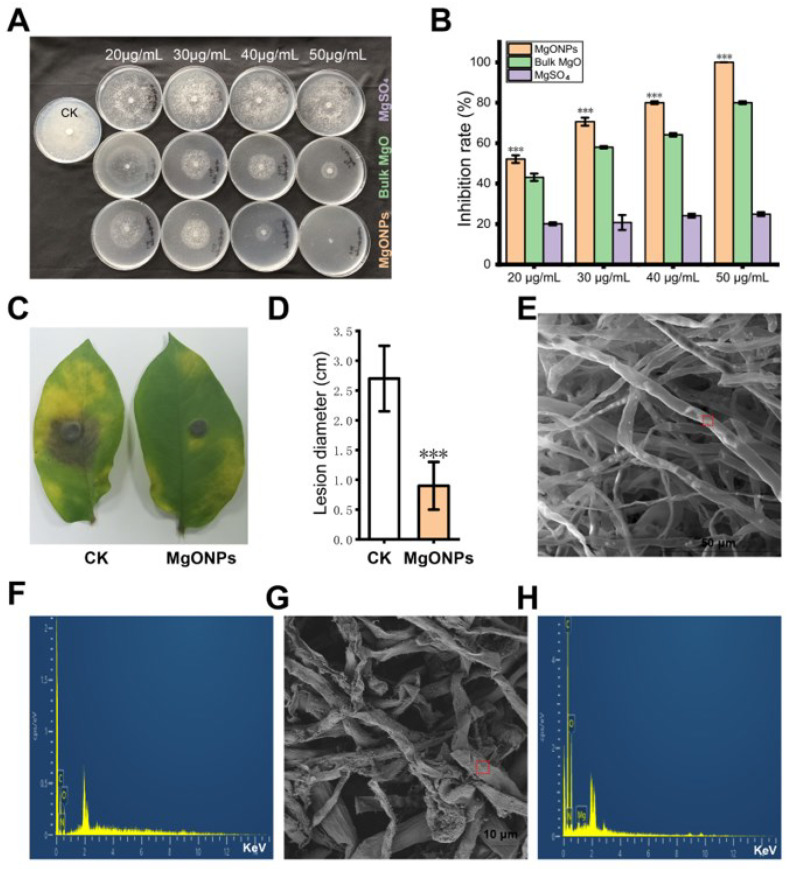
The antimicrobial activity of the MgSO_4_, bulk MgO, and MgO NP against *P. infestans*. (**A**) Comparison of the mycelial growth of *P. infestans* under MgSO_4_, bulk MgO and MgO NPs treatment. The photograph was taken after 7 days of culture in the dish. (**B**) The inhibition rate statistics of the MgSO_4_, bulk MgO and MgO NPs against *P. infestans*. (**C**) Comparison of the infection disease spot of potato blight after spraying MgO NPs or sterile water (CK) on the potato leaves. (**D**) The plant disease spot diameter statistics of potato blight after spraying MgO NPs or sterile water (CK). *** represents significant difference at *p* < 0.01. (**E**,**G**) Scanning electron microscopy observations of oomycete hyphal morphological changes directly exposed to sterile water (**E**) or MgO NPs, respectively (**G**). The red frame is the area for energy-dispersive spectrometry. (**F**,**H**) The corresponding energy-dispersive spectrometry data represent the qualitative elemental analysis of MgO NPs for using the red boxes in (**E**) and (**G**), respectively.

**Figure 3 toxics-10-00553-f003:**
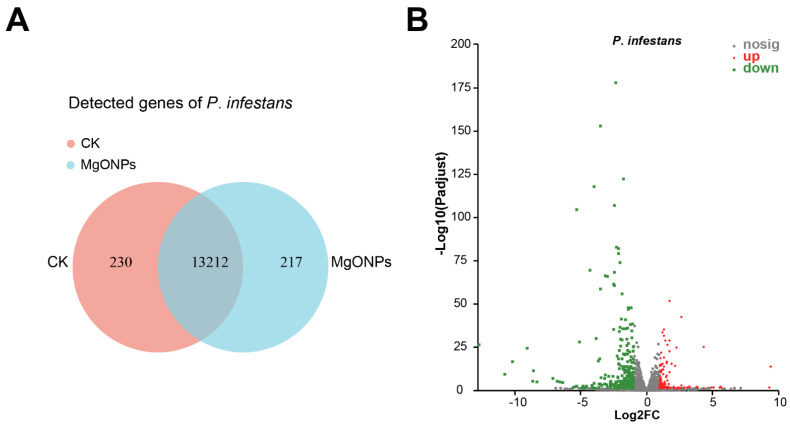
Transcriptome profiles of *P. infestans* after 30 min MgO NP exposure. (**A**) Venn diagram showing the number of expressed genes in *P. infestans* upon MgO NP exposure and control. (**B**) Volcano plot showing DEGs of *P. infestans* after 30 min MgO NP exposure. Red dots indicate up-regulated genes, green dots indicate down-regulated genes, and grey dots represent no significantly differentially expressed genes. The X-axis shows the fold-change values between the control and MgO NP exposure groups, based on a log_2_ scale, and the Y-axis shows the P-adjust value of differentially expressed genes based on a log_10_ scale.

**Figure 4 toxics-10-00553-f004:**
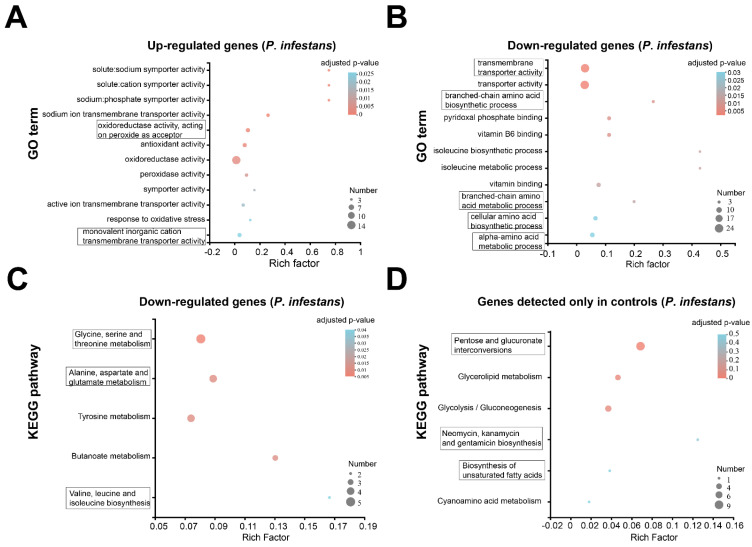
GO and KEGG analysis for the transcriptomic data of *P. infestans* after MgO NP exposure. (**A**,**B**) GO enrichment analysis of DUGs (**A**) and DDGs (**B**), respectively. (**C**,**D**) KEGG pathway enrichment analysis of DUGs (**C**) and control-only expressed genes (**D**), respectively. Enriched GO terms or KEGG pathways are shown in a bubble diagram. The horizontal axis shows the Rich factor (the ratio between the number of concerned genes and the number of total transcription genes in the corresponding term). The vertical coordinates indicate the different GO terms or KEGG pathways. The size of the point shows that concerned gene numbers in corresponding GO term or KEGG pathways. The color of each bubble indicates the adjusted *p*-value.

**Figure 5 toxics-10-00553-f005:**
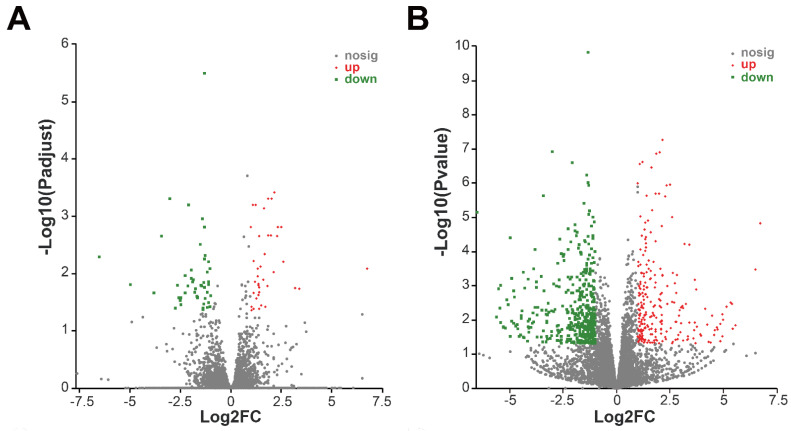
Transcriptome profiles of potato after 6 h MgO NP exposure. (**A**) Volcano plot showing DEGs of potato after MgO NP exposure. The adjusted *p*-value < 0.05 and fold changes <0.5 or >2.0 were used as the cutoff. (**B**) Volcano plot showing DEGs of potato after MgO NP exposure. The *p*-value < 0.05 and fold changes <0.5 or >2.0 were used as cutoff. Red dots indicate up-regulated genes, and green dots indicate down-regulated genes, grey dots represent no significantly different expressed genes. The X-axis shows the fold-change values between the control and MgO NP exposure groups, based on a log_2_ scale, and the Y-axis shows the adjusted *p*-value or *p*-value of differentially expressed genes based on a log_10_ scale.

**Figure 6 toxics-10-00553-f006:**
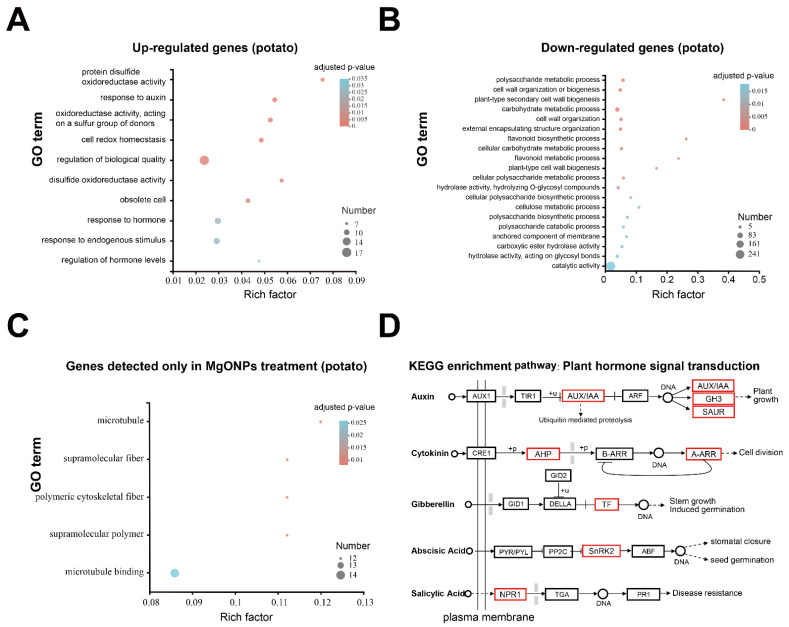
GO and KEGG analysis for the transcriptomic data of potato after 6 h of MgO NP exposure. (**A**–**C**) GO enrichment analysis of DUGs, (**A**) DDGs, (**B**) and genes, which were only detected in MgO NP treatment (**C**). Enriched GO terms are shown in a bubble diagram. The horizontal axis shows the Rich factor (the ratio between the number of concerned genes and the number of total transcription genes in the corresponding term). The vertical coordinates indicate the different GO terms. The size of the point shows the concerned gene numbers and their corresponding GO terms. The color of each bubble indicates the adjusted *p*-value. (**D**) The KEGG term of “plant hormone signal transduction” was significantly enriched in DUGs. The red boxes represent genes that are up-regulated.

**Figure 7 toxics-10-00553-f007:**
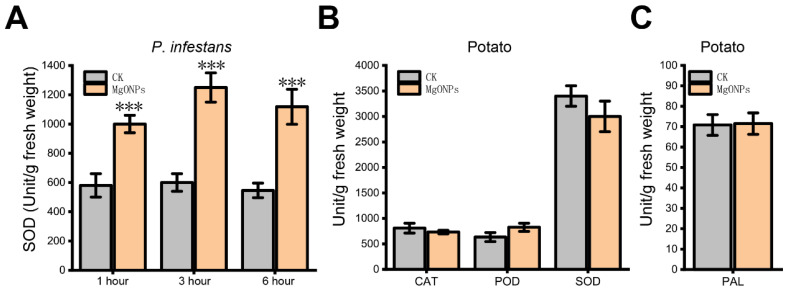
Effect of the MgO NPs on the antioxidant system and PAL activity. (**A**) SOD activities of *P. infestans* after 1, 3, and 6 h of MgO NPs or sterile water (CK) exposure. (**B**) CAT, POD, and POD activities of potato after 6 h of MgO NPs or sterile water (CK) exposure. (**C**) PAL activity of potato after 6 h of MgO NPs or sterile water (CK) exposure. Error bars represent the standard deviation, *** *p* < 0.01.

## Data Availability

Raw sequencing data for *P*. *infestans* and potato were deposited in the National Genomics Data Center Genome Sequence Archive (NGDC GSA) under the accession number CRA007754 and CRA007804, respectively.

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
