# Peer review of "Toxicity Effects and Mechanisms of MgO Nanoparticles on the Oomycete Pathogen Phytophthora infestans and Its Host Solanum tuberosum"

_toxics, 2022, doi:10.3390/toxics10100553_

Round 1
Reviewer 1 Report
In this study, the authors assessed the direct toxicity activity of MgONPs on P. infestans and the protection ability of potatoes against P. infestans. And then, the destruction of P. infestans hyphae by MgONPs was confirmed through a scanning electron microscope (SEM). The language should be revised. Also, some new studies must add to enrich the paper.
-what is the novelty of the current study? There are many similar papers in this field
-there are some concerns about the toxicity of inorganic NPS. please address this issue in the introduction.
-please cite the following papers to enrich the introduction with new studies: https://doi.org/10.1016/j.envpol.2022.119606, http://doi.org/10.26480/mjsa.01.2021.16.20, https://doi.org/10.1016/j.scitotenv.2022.154058, https://doi.org/10.1007/s10725-021-00782-w
-how samples were prepared for SEM and TEM analysis
-also what was the angle and temperature of DLS? The detailed methodology was expected
-the XRD analysis didn’t add in the “methods” section
-the crystalline size of NPs should be calculated from the XRD pattern
-how did you claim that the “XRD pattern indicated that there was high purity”? please explain with your data
-Figures 4 and 6 have low resolution
Author Response
Response to Reviewer 1 Comments
In this study, the authors assessed the direct toxicity activity of MgONPs on P. infestans and the protection ability of potatoes against P. infestans. And then, the destruction of P. infestans hyphae by MgONPs was confirmed through a scanning electron microscope (SEM). The language should be revised. Also, some new studies must add to enrich the paper.
Point 1: what is the novelty of the current study? There are many similar papers in this field.
Response 1: Thanks for your question. Recently, nanotechnology is increasingly exploited in a wide range of agricultural application and provides a novel way to manage plant disease for increasing food production in a sustainable manner. Many inorganic and organic nanomaterials have been found to exhibit excellent antibacterial, antifungal, and antiviral properties on phytopathogenic microbes in laboratory and even field conditions. However, both toxicity mechanisms of nanoparticles on target pathogens and molecular modulation of nanoparticles on protected plants are still largely unclear. As most of previous studies focused on material characteristics of nanoparticles to elucidate their antimicrobial mechanisms, the molecular responses of pathogen or host plant were largely unknown although these molecular responses were essential to understand biological changes of organisms under nanoparticles’ exposure and the toxic mechanisms of nanoparticles. Magnesium oxide nanoparticles (MgO NPs), which were considered as biocompatibile, safe and inexpensive nanoparticles, were synthesised and characterized by a green way in our study as previous study. Here, we newly found that MgO NPs could protect potato against P. infestans at a low dosage (50 μg/mL), firstly displayed molecular response of P. infestans and potato under MgONPs treatment, firstly illustrated the toxic mechanisms of MgO NPs on P. infestans, and firstly found the potentially novel benefit effects by spraying MgO NPs on potato except fertilizer supplying. The toxic mechanisms of MgO NPs on P. infestan include the continuous ROS stress, cell surface disruption, and inhibition of both membrane transport capacity and essential metabolic pathways on P. infestans. Compared with other researches about the toxic mechanism of metal oxide nanoparticles, our study found that the inhibition of ten essential metabolic pathways is one of the toxic mechanisms of MgO NPs and the dissolved magnesium ions does not play a major role in the toxicity of MgO NPs. On the other hand, we also found that MgO NPs not only had no obvious harmful effects on potato, but also potentially modulate plant hermone signaling pathway to promote plant growth and disease resistance. Taken together, our study will not only provide an potentially effective strategy for preventing potato late blight in agricultural application, but also extend the understanding of toxic mechanisms and effects of NPs in agricultural system.
Point 2: there are some concerns about the toxicity of inorganic NPS. please address this issue in the introduction.
Response 2: Thank you very much for this kind suggestion. We have addressed this issue in the introduction section at the third paragraph in our revised manuscript.
Point 3: please cite the following papers to enrich the introduction with new studies: https://doi.org/10.1016/j.envpol.2022.119606, http://doi.org/10.26480/mjsa.01.2021.16.20, https://doi.org/10.1016/j.scitotenv.2022.154058, https://doi.org/10.1007/s10725-021-00782-w
Response 2: Thank you for your kind suggestion. According your suggestions, we have cited the papers of “https://doi.org/10.1016/j.envpol.2022.119606” and “https://doi.org/10.1007/s10725-021-00782-w” in our revised manscript. These two cited papers were at ref 8 and ref 24, respectively. The pappers from “http://doi.org/10.26480/mjsa.01.2021.16.20” and “https://doi.org/10.1016/j.scitotenv.2022.154058” were entitled “Promising early planting and stress-tolerant potato genotypes for northern Bangladesh” and “Combined antimicrobial effect of bacteriocins with other hurdles of physicochemic and microbiome to prolong shelf life of food: A review”, respectively. The contents of these two articles were completely unrelated with our study or researches about NPs. You might have a copy error in the process of reviewing our manuscript. We did not cite these two articles in our manuscript.
Point 4: how samples were prepared for SEM and TEM analysis
Response 4: Thanks for your questions. MgO NPs were dispersed in ultrapure water, and spread by using an ultrasonic bath sonicator before scanning electron microscope (SEM) and transmission electron microscopy (TEM). The morphology and agglomeration of MgO NPs were visualized by SEM with ZEISS Sigma 300 (Germany) after the samples were conductively coated by gold sputter (<10 nm). The crystalline morphology and struc-ture of MgO NPs were determined by using TEM/HRTEM on a FEI Tecnai G2 F20 (America) Scanning Transmission Electron Microscope with an accelerating volt-age of 300 kV. There were different preparation methods for NPs and biological sample of P. infestans before SEM. For the SEM of P. infestans, the fresh P. infestans hyphae cubes were incubated cultured on the RSA medium plats, which were coated with sterilized cellophane. After mycelia covered most of the area on the petri dishes, P. infestans were sprayed with MgO NPs (50mg/L) or sterile water. After 6 hours later, mycelia of P. infestans were collected, gently washed with PBS solution (pH 7.4) and incubated with 2.5% glutaraldehyde 4 hours. Subsequently, the hyphal samples were then dehydrated by using a series of concentrations (30–100%) of ethanol. After being air-dried naturally, the samples were conductively coated by gold sputter (<10 nm) and submitted to SEM (FEI Quanta 200, Netherlands) operated at accelerating voltages at 30 kV. We have reedited and improved the section of “Materials and Methods” in our revised manuscript according suggestions from you and other reviewers. Thanks again.
Point 5: also what was the angle and temperature of DLS? The detailed methodology was expected
Response 5: Thanks for your question. There was no experiment of Zetasizer Dynamic Light Scattering (DLS) to detect the size of NPs in our study for that the agglomeration of our MgO NPs could result a incorrect size. Thus, the detailed methodology and resut of DLS were all not suppled in our study.
Point 6: the XRD analysis didn’t add in the “methods” section
Response 6: Thanks for your reminder. We are very soory for this cursoriness and we have added the method of XRD in our revised manuscript at the section of 2.1.
Point 7: the crystalline size of NPs should be calculated from the XRD pattern
Response 7: Thanks for your suggestion. By analyzing the XRD peak pattern, The crystalline sizes of NPs are 85.6 nm, 77.9nm, 62.5nm, 65nm and 69.6nm, respec-tively. We also revised this mistake in our manuscript.
Point 8: how did you claim that the “XRD pattern indicated that there was high purity”? please explain with your data
Response 8: Thanks for your question. No other peaks were detected in the XRD spectrum, indicating the high purity of the obtained MgO NPs. We also revised our sentence about the description on this result in the place of the original manuscript.
Point 9: Figures 4 and 6 have low resolution
Response 9: Thanks for your question. Figure 4 and 6 were all made up of different vector images, which were calculated by computer. The resolutions of Figure 4 and 6 are all 600dpi. The lack of sensory clarity may be due to that some fonts are not pure black in the picture. We also modified this and provided new Figures in our manuscript. We also have provided separate images for our manuscript with high resolution and definition. On the other hand, image quality might be also affected by Word document file. We also suggested that you can visulized separate images of Figures 4 and 6.

Reviewer 2 Report
The work by wang et al. (Toxicity effects and mechanisms of MgO nanoparticles on the oomycete pathogen Phytophthora infestans and its host potato) reports the protective potential of magnesium oxide nanoparticles (MgO NPs) for potato against Phytophthora infestans (P. infestans) at a low dosage (50 μg/mL). For this, the authors performed NP fabrication and characterization, P. infestans and potato culture, antimicrobial activity assay of MgONPs to P. Ä°nfestans, SOD, POD, CAT and PAL activity assays, RNA extraction and sequencing, RNA isolation, cDNA synthesis and quantitative RT‑PCR. I think the manuscript provides valuable information to this research field. However, I noticed some major issues that must be eliminated before the publication.
I think it would be better to use the abbreviation of nanoparticles as MgO-NPs or MgO NPs instead of MgONPs.
The experimental section must be rewritten. In the pesent form, it is highly difficult to understand the procedures.
The detailed SEM analysis must be performed to show the presence of the Phytophthora infestans and MgO-NPs onto the infected potato leaves. The authors must provide these structures at different magnifications. Interestingly, only oomycete hyphal parts of P. infestans was shown in Fig.2E-F. The presence of P. infestans must be confirmed in detail. Also, the Fig.2E-F does not support the the authors’ claim (collapse of tube-like hyphae structure). Again, many SEM images are required to confirm the evaluation. This data is essential for the reliability of the report.
Please provide legend and unit for y-axis of Fig.2D.
Line 253-254 Please fix the figure numbers as Figure 2F and Figure 2G in text.
For EDX data, the data for the negative control must be represented for the comparison.
The conclusion part is missing in the text. Please summarize the general findings of the report in this section.
I noticed many grammatical errors that make it so difficult to understand the manuscript. Some statements are too long. Please fix these issues.
Author Response
Response to Reviewer 2 Comments
The work by wang et al. (Toxicity effects and mechanisms of MgO nanoparticles on the oomycete pathogen Phytophthora infestans and its host potato) reports the protective potential of magnesium oxide nanoparticles (MgO NPs) for potato against Phytophthora infestans (P. infestans) at a low dosage (50 μg/mL). For this, the authors performed NP fabrication and characterization, P. infestans and potato culture, antimicrobial activity assay of MgONPs to P. Ä°nfestans, SOD, POD, CAT and PAL activity assays, RNA extraction and sequencing, RNA isolation, cDNA synthesis and quantitative RT‑PCR. I think the manuscript provides valuable information to this research field. However, I noticed some major issues that must be eliminated before the publication.
Point 1: I think it would be better to use the abbreviation of nanoparticles as MgO-NPs or MgO NPs instead of MgONPs.
Response 1: Thanks for your kind suggestion. MgO NPs was used as the abbreviation of magnesium oxide nanoparticles in our revised manuscript.
Point 2: The experimental section must be rewritten. In the present form, it is highly difficult to understand the procedures.
Response 2: Thanks for your kind suggestion. We have rewritten the experimental section in our revised manuscript. And then, our revised manuscript has been polished by a native English-speaking colleague through https://www.mdpi.com/authors/english (English Editing ID: english-50065). Thanks again.
Point 3: The detailed SEM analysis must be performed to show the presence of the Phytophthora infestans and MgO-NPs onto the infected potato leaves. The authors must provide these structures at different magnifications. Interestingly, only oomycete hyphal parts of P. infestans was shown in Fig.2E-F. The presence of P. infestans must be confirmed in detail. Also, the Fig.2E-F does not support the authors’ claim (collapse of tube-like hyphae structure). Again, many SEM images are required to confirm the evaluation. This data is essential for the reliability of the report.
Response 3: Thanks for your suggestions. To characterize the toxic effects of the MgO NPs attachment on hyphae of P. infestans, the morphological changes of P. infestans cells by MgO NPs were further monitored through biological SEM in our study. The infected potato leaves were not used to observe the morphological changes of P. infestans through SEM, for that many factors, incuding nutrition, infection stage and other microbials on leaf surface, could affect the hyphae morphology of mycelium on leaf surface. To reduce potential influence factors, fresh mycelia of P. infestans were sprayed with MgO NPs or sterile water and visulized by SEM. We provided other SEM figures with bigger visual field and EDX data to the reliability of our results in our revised manuscript. Here, we also provide two other SEM figures with big visual field to you. Thanks again.
Point 4: Please provide legend and unit for y-axis of Fig.2D.
Response 4: Thanks for your kind reminder. We have provided legend and unit for y-axis of Fig.2D in our revised manuscript.
Point 5: Line 253-254 Please fix the figure numbers as Figure 2F and Figure 2G in text.
Response 5: Thanks for your kind reminder. We have modified this mistake in our revised manuscript.
Point 6: For EDX data, the data for the negative control must be represented for the comparison.
Response 6: Thanks for your kind reminder. We have the negative controls in previous EDX data. Considering the space for pictures is limited, we omited this negative control in previous manuscript. Here, we narrowed the picture and provided the EDX data on negative control in our revised manuscript.
Point 7: The conclusion part is missing in the text. Please summarize the general findings of the report in this section.
Response 7: Thanks for your kind suggestion. We had no conclusion part in previous manuscript for that the section of conclusion is not mandatory according the journal requirements. We have added the “concusion” section in our revised manuscript according the suggestion from you and other reviewers.
Point 8: I noticed many grammatical errors that make it so difficult to understand the manuscript. Some statements are too long. Please fix these issues.
Response 8: Thanks for your kind suggestion. Our revised manuscript has been polished by a native English-speaking colleague through https://www.mdpi.com/authors/english (English Editing ID: english-50065).

Reviewer 3 Report
Comments on the MSS: Toxicity effects and mechanisms of MgO nanoparticles on the oomycete pathogen Phytophthora infestans and its host potato
Comment to the Author: The authors presented a good approach in the field of applications of green synthesised MgONPs in the protection of potato against Phytophthora infestans and evaluated its toxicity effect on the host plants. The application of MgONPs as an antifungal agent is a very recently come into light in the researcher community and will be a promising approach in this area of research. The authors provided the good evidences to prove the study and given the detailed data in MS and as supplementary material. The manuscript is well written by the authors but needs minor corrections in its current form.
The following are some points related with the manuscript. The manuscript could be accepted after minor revision-
1. It could better reflect title if use of generic name for “Potato” in title.
2. Line 37 ‘P. infestans, one of the important pathogens in the agriculture, infected on the potato or ....’ should check the sentence construction. It may be ‘infected to the..’
3. Authors can use “NPs’ in place of ‘Nanoparticles’ throughout the MS after its first mention.
4. Line 53, ‘..such as TiO2 [8], CuO [9], 53 ZnO [10], AgO [11] and MnO2 [12] nanomaterials..’, check for subscript of valences.
5. Line 54 ‘..against to phytopathogenic..’ it should be ‘‘..against phytopathogenic..’ remove ‘to’ from sentence.
6. Line 58, ‘..metal-based nanopartciles have been..’ check spelling for nanoparticles.
7. Line 71-74, ‘As most of previous studies.….on different organisms’ is a very long, difficult to understand and unclear sentence, revise and rephrase the sentence for proper understanding and clear meaning.
8. Line 84 ‘cell death on bacterial cell [26]’, please check?
9. Line 102 ‘DEGs with loosed threhold (p-value < 0.05..’ should check for spelling.
10. Line 105 ‘phenylalnine ammonialyase (PAL) suggested’, again check spelling.
11. Line 119 ‘Mg(NO3)2 and ammonia solution were mixed at room’ numbers in the formula should be in subscript.
12. Line 120 ‘Mg2+ and OH− ions..’ valence should be in superscript.
13. Line 123 ‘Mg(OH)2 to nano-MgO’ subscript ‘2’
14. Line 131-133 ‘The potato cultivar “FA-131 VORITA” was routinely cultured at 22℃ in the green house with 16 h of light and 8 h of dark daily’, how one can adjust these conditions in green house? Please specify is it green house or growth chamber?
15. Line 135 ‘For the evaluation of the antimicrobial activity of MgONPs in vitro’, authors may rephrase the sentence ‘For the in vitro antimicrobial activity evaluation of MgONPs’. In vitro should be italicized.
16. Line 143 ‘To evaluate the protection activity of MgONPs to P. infestans in vivo ‘, authors may rephrase the sentence ‘To evaluate the in vivo protection activity of MgONPs to P. infestans’. In vivo should be italicized.
17. Line 151 ‘RSA plats’ what is it? Give brief explanation.
18. Line 215 ‘minimize the gap between heading and text matter in results’.
19. Authors should check the spellings, grammar, spacing between the overlapping words, punctuation, etc. throughout the text, as there are some incomplete words, mistakes in punctuations, etc.
20. Conclusion section is missing in the MS.
21. Journal names are missing in almost all the Reference for eg. refs. 1-5, 27, 29.
22. References should be checked for punctuations, italicized, formatting of the titles, bold formatting for year, and as per journal format, etc.
Review comments:
The manuscript could be accepted as after major revision.
Author Response
Response to Reviewer 3 Comments
Comment to the Author: The authors presented a good approach in the field of applications of green synthesised MgONPs in the protection of potato against Phytophthora infestans and evaluated its toxicity effect on the host plants. The application of MgONPs as an antifungal agent is a very recently come into light in the researcher community and will be a promising approach in this area of research. The authors provided the good evidences to prove the study and given the detailed data in MS and as supplementary material. The manuscript is well written by the authors but needs minor corrections in its current form.
The following are some points related with the manuscript. The manuscript could be accepted after minor revision-
- It could better reflect title if use of generic name for “Potato” in title.
Response 1: Thanks for your kind suggestion. We have revised our titile.
- Line 37 ‘P. infestans, one of the important pathogens in the agriculture, infected on the potato or ....’ should check the sentence construction. It may be ‘infected to the..’
Response 2: Thanks for your kind suggestion. We have revised this mistake in our manuscript.
- Authors can use “NPs’ in place of ‘Nanoparticles’ throughout the MS after its first mention.
Response 3: Thanks for your kind suggestion. ‘NPs’ was used to in place of ‘Nanoparticles’ in our manuscript.
- Line 53, ‘..such as TiO2 [8], CuO [9], 53 ZnO [10], AgO [11] and MnO2 [12] nanomaterials..’, check for subscript of valences.
Response 4: Thanks for your kind suggestion. We have checked the subscript of valences and revised them in our manuscript.
- Line 54 ‘..against to phytopathogenic..’ it should be ‘‘..against phytopathogenic..’ remove ‘to’ from sentence.
Response 5: Thanks for your kind suggestion. We revised this mistake in our manuscript.
- Line 58, ‘..metal-based nanopartciles have been..’ check spelling for nanoparticles.
Response 6: Thanks for your kind suggestion. We revised this mistake in our manuscript.
- Line 71-74, ‘As most of previous studies.….on different organisms’ is a very long, difficult to understand and unclear sentence, revise and rephrase the sentence for proper understanding and clear meaning.
Response 7: Thanks for your kind suggestion. We have modified this sentence according a native English-speaking colleague from https://www.mdpi.com/authors/english (English Editing ID: english-50065).
- Line 84 ‘cell death on bacterial cell [26]’, please check?
Response 8: Thanks for your kind suggestion. It should be ‘The bactericide activity of MgO NPs further reported that MgO NPs could cause cell membrane leakage, induces oxidative stress and ultimately leads to bacterial cell death’.
- Line 102 ‘DEGs with loosed threhold (p-value < 0.05..’ should check for spelling.
Response 9: Thanks for your kind suggestion. We have removed this sentence in our revised manuscript according other reviewers.
- Line 105 ‘phenylalnine ammonialyase (PAL) suggested’, again check spelling.
Response 10: Thanks for your kind suggestion. We have revised this mistake in our manuscript.
- Line 119 ‘Mg(NO3)2 and ammonia solution were mixed at room’ numbers in the formula should be in subscript.
Response 11: Thanks for your kind suggestion. We have revised this mistake in our manuscript.
- Line 120 ‘Mg2+ and OH− ions..’ valence should be in superscript.
Response 12: Thanks for your kind suggestion. We have revised this mistake in our manuscript.
- Line 123 ‘Mg(OH)2 to nano-MgO’ subscript ‘2’
Response 13: Thanks for your kind suggestion. We have revised this mistake in our manuscript.
- Line 131-133 ‘The potato cultivar “FA-131 VORITA” was routinely cultured at 22℃ in the green house with 16 h of light and 8 h of dark daily’, how one can adjust these conditions in green house? Please specify is it green house or growth chamber?
Response 14: Thanks for your kind question. We are soory for our mistake in the word’s meaning. The potato cultivar “FAVORITA” was routinely cultured at 22℃ in the plant growth chambergreen house with 16 h of light (8000-10000 lux) and 8 h of dark daily
- Line 135 ‘For the evaluation of the antimicrobial activity of MgONPs in vitro’, authors may rephrase the sentence ‘For the in vitro antimicrobial activity evaluation of MgONPs’. In vitro should be italicized.
Response 15: Thanks for your kind suggestion. The sentence has been revised as ‘To evaluate the in vitro antimicrobial activity of MgO NPs,…’
- Line 143 ‘To evaluate the protection activity of MgONPs to P. infestans in vivo ‘, authors may rephrase the sentence ‘To evaluate the in vivo protection activity of MgONPs to P. infestans’. In vivo should be italicized.
Response 16: Thanks for your kind suggestion. We have revised this according your suggestion.
- Line 151 ‘RSA plats’ what is it? Give brief explanation.
Response 17: Thanks for your kind question. RSA is the abbreviation of ‘rye sucrose agar’ and we have added this explanation in our revised manuscript.
- Line 215 ‘minimize the gap between heading and text matter in results’.
Response 18: Thanks for your kind suggestion. We writted our manuscript on the template of Toxics and the format was set in advance.
- Authors should check the spellings, grammar, spacing between the overlapping words, punctuation, etc. throughout the text, as there are some incomplete words, mistakes in punctuations, etc.
Response 19: Thanks for your kind suggestion. Our revised manuscript has been polished by a native English-speaking colleague at the website of ‘https://www.mdpi.com/authors/english’ (English Editing ID: english-50065).
- Conclusion section is missing in the MS.
Response 20: Thanks for your kind suggestion. We had no conclusion part in previous manuscript for that the section of conclusion is not mandatory according the journal requirements. We have added the ‘concusion’ section in our revised manuscript according the suggestion from you and other reviewers. Thanks again.
- Journal names are missing in almost all the Reference for eg. refs. 1-5, 27, 29.
Response 21: Thanks for your kind suggestion. We have revised these mistakes in our revised manuscript.
- References should be checked for punctuations, italicized, formatting of the titles, bold formatting for year, and as per journal format, etc.
Response 22: Thanks for your kind suggestion. We have checked these in our revised manuscript according your kind suggestion.
Review comments:
The manuscript could be accepted as after major revision.

Reviewer 4 Report
The work sent for review is very interesting and presents the possibilities of the practical application of nanoparticles in agriculture to protect potato against Phytophthora infestans. The toxicity mechanism of MgONPs to P. infestans were explained through microscopic observation, antioxidant activity measurements, and transcriptomic analysis. It has appeared that NPs are an effective strategy for preventing phytopathogen infections. The authors described ten metabolism pathways stunted by MgONPs, which is a novel discovery in toxic mechanisms of metal-base nanoparticles. For this reason, this report is important and should be published. As a reviewer, however, I would like to draw your attention to some doubts concerning the above report.
It is known that the activity of nanoparticles is closely related to their shape and size. The authors independently synthesized NP and obtained a product that is rather heterogeneous in size and shape. The authors wrote that the hydrodynamic diameters were measured with the Malvern Zetasizer Nano Series. It would be good to give the zeta potential and the hydrodynamic diameter of the biosynthesized particles.
Minor mistakes:
Introduction
l.93-115- Please do not describe your results here.
l.132- please specified light qualities and light frequency (lux).
Discusion
Subchapter 4.1. should be transferred to results.
The authors compared their results to paper 14. Chen, J.; Wu, L.; Lu, M.; Lu, S.; Li, Z.Y.; Ding, W.J.F.i.M. Comparative Study on the Fungicidal Activity of Metallic MgO Nano- 612 particles and Macroscale MgO Against Soilborne Fungal Phytopathogens. 2020, 11, 365).
Please compare the effect in the context of NPs characteristics.
The is a lack of conclusion-please complete.
Author Response
Response to Reviewer 4 Comments
The work sent for review is very interesting and presents the possibilities of the practical application of nanoparticles in agriculture to protect potato against Phytophthora infestans. The toxicity mechanism of MgONPs to P. infestans were explained through microscopic observation, antioxidant activity measurements, and transcriptomic analysis. It has appeared that NPs are an effective strategy for preventing phytopathogen infections. The authors described ten metabolism pathways stunted by MgONPs, which is a novel discovery in toxic mechanisms of metal-base nanoparticles. For this reason, this report is important and should be published. As a reviewer, however, I would like to draw your attention to some doubts concerning the above report.
Point 1: It is known that the activity of nanoparticles is closely related to their shape and size. The authors independently synthesized NP and obtained a product that is rather heterogeneous in size and shape. The authors wrote that the hydrodynamic diameters were measured with the Malvern Zetasizer Nano Series. It would be good to give the zeta potential and the hydrodynamic diameter of the biosynthesized particles.
Response 1: Thanks for your sugestion. We are very sorry for this mistak. The zeta potential was added in our manuscript. Considering the agglomeration of MgO NPs could infulence the measurement of the hydrodynamic diameter, the size of synthesized NP was measured by TEM. The average size of MgO NPs was calculated by the TEM for about 100 NPs. Thus, we just added the zeta potential of MgO NPs in our revised manuscript.
Minor mistakes:
Introduction
Point 2: l.93-115- Please do not describe your results here.
Response 2: Thanks for your suggestion. We have rewritten this paragraph in our revised manuscript.
Point 3: l.132- please specified light qualities and light frequency (lux).
Response 3: Thanks for your kind suggestion. The potato cultivar “FAVORITA” was routinely cultured at 22℃ in the green house with 16 h of light with 8000-10000 lux and 8 h of dark daily. We have refined the information in our revised manuscript. Thanks again.
Discusion
Point 4: Subchapter 4.1. should be transferred to results.
Response 4: Thanks for your sugestion. Subchapter 4.1. has been rewritten to discussed the results of the MgO NPs’ synthesis and its characteristics.
Point 5: The authors compared their results to paper 14. Chen, J.; Wu, L.; Lu, M.; Lu, S.; Li, Z.Y.; Ding, W.J.F.i.M. Comparative Study on the Fungicidal Activity of Metallic MgO Nano- 612 particles and Macroscale MgO Against Soilborne Fungal Phytopathogens. 2020, 11, 365). Please compare the effect in the context of NPs characteristics.
Response 5: Thanks for your sugestions. Considering there were different pathogens between our study and the study of Chen et. al, it was irrational to compare the toxic effect (dosage) of MgO NOS between our study and the study of Chen et. al. Thus, we providede possible reason of the high toxicty (low dosage) of our MgO NPs in our discussion. The possible reason was that MgO NPs had higher toxicity than previous MgO NPs or MgO NPs was more applicable to inhibit P. infestans than P. parasitica. The morphological characteristics of our MgO NPs was irregularly spherical with some sliced shells. This sliced shells, which were not present in the MgO NPs from Chen et. al., may lead high antimicrobial activities. These discussion was added at the discussion of NPs characterristics (4.1) in our revised manuscript.
Point 6: There is a lack of conclusion-please complete.
Response 6: Thanks for your kind suggestion. We had no conclusion part in previous manuscript for that the section of conclusion is not mandatory according the journal requirements. We have added the “concusion” section in our revised manuscript according the suggestion from you and other reviewers. Thanks again.

Round 2
Reviewer 1 Report
the revised version of the paper can be accepted
Author Response
Thanks very much for your kind suggestions and patience.
Reviewer 2 Report
I think the report can be published in its present form.
Author Response

(The authors gave the same response as above.)

Reviewer 3 Report
Comment to the Author: The revised manuscript is now ok but needs some more minor corrections in its current form.
The following are some points related with the manuscript. The manuscript could be accepted after minor revision-
1. Check the overlapping of the words throughout the MS for eg. in 50mg/L, give the space between number and unit ‘50 mg/L’ like this. Do for all.
2. Also space between the text and the [] bracket while citation throughout the MS check carefully.
3. Line 195, spacing between symbols and words ‘(OD260/280=1.8~2.0, OD260/230≥2.0, RIN≥6.5, 28S:18S≥1.0, 195 ≥100ng/μl, ≥2μg)’
4. Line 238-240 ‘The XRD pattern (Figure 1D) showed that several sharp peaks, which were located at 2" of 36.95 42.92, 62.30, 239 74.76, and 78.61, were assigned to the (111), (200), (220), (311), and (222) crystallographic’ check the sentence for addition of symbols.
5. Line 244, ‘The crystalline sizes of NPs are 85.6 nm, 77.9nm, 62.5nm, 65nm and 69.6nm, respectively’ give space between overlapping words.
6. Describe ‘DEGs’ when it appears first. Then use its abbreviation.
7. Figure 2 caption should be checked for superscript of ‘4’ in ‘MgSO4’
8. Line 259, checked for superscript of ‘4’ in ‘MgSO4’
9. Figure 4 A,B,C check the spelling for ‘regulated’ and correct it in the figure.
10. Figure 5 Check ‘Log10’ and ‘Log2’ for superscripts, also spacing between overlapping words ‘P value’ and ‘P adjust’
11. Line 473, give the spacing in overlapping word ‘(50mg/L)’
12. Check the reference section. It appears the repetition of numbering from ref. no 48 and check in the text accordingly.
Author Response
Comment to the Author: The revised manuscript is now ok but needs some more minor corrections in its current form.
The following are some points related with the manuscript. The manuscript could be accepted after minor revision-
- Check the overlapping of the words throughout the MS for eg. in 50mg/L, give the space between number and unit ‘50 mg/L’ like this. Do for all.
Response 1: Thanks for your kind suggestion. We have given the space between number and unit.
- Also space between the text and the [] bracket while citation throughout the MS check carefully.
Response 2: Thanks for your kind suggestion. We have given the space between the text and the [] bracket.
- Line 195, spacing between symbols and words ‘(OD260/280=1.8~2.0, OD260/230≥2.0, RIN≥6.5, 28S:18S≥1.0, 195 ≥100ng/μl, ≥2μg)’
Response 3: Thanks for your kind suggestion. We have given the space between symbols and words. Additionally, we have also given space between number and unit, and added words before ‘≥100ng/μl’ and ‘≥2μg’.
- Line 238-240 ‘The XRD pattern (Figure 1D) showed that several sharp peaks, which were located at 2" of 36.95 42.92, 62.30, 239 74.76, and 78.61, were assigned to the (111), (200), (220), (311), and (222) crystallographic’ check the sentence for addition of symbols.
Response 4: Thanks for your kind suggestion. We have revised this sentence and added some essential symbols.
- Line 244, ‘The crystalline sizes of NPs are 85.6 nm, 77.9nm, 62.5nm, 65nm and 69.6nm, respectively’ give space between overlapping words.
Response 5: Thanks for your kind suggestion. We have given the space between number and unit.
- Describe ‘DEGs’ when it appears first. Then use its abbreviation.
Response 6: Thanks for your kind suggestion. DEGs is the abbreviation of ‘differentially expressed genes’ and we have added this explanation in our revised manuscript when it appeared first.
- Figure 2 caption should be checked for superscript of ‘4’ in ‘MgSO4’
Response 7: Thanks for your kind suggestion. The superscript of ‘4’ in ‘MgSO4 has been changed in our revised manuscript, including text and figure.
- Line 259, checked for superscript of ‘4’ in ‘MgSO4’
Response 8: Thanks for your kind suggestion. The superscript of ‘4’ in ‘MgSO4 has been changed in our revised manuscript.
- Figure 4 A,B,C check the spelling for ‘regulated’ and correct it in the figure.
Response 8: Thanks for your kind suggestion. We have revised this spelling mistake.
- Figure 5 Check ‘Log10’ and ‘Log2’ for superscripts, also spacing between overlapping words ‘P value’ and ‘P adjust’
Response 8: Thanks for your kind suggestion. The superscripts of ‘Log10’ and ‘Log2’ have been revised. ‘P-value’ and ‘P-adjust’ have been changed as ‘p-value’ and ‘adjusted p-value’, respectively.
- Line 473, give the spacing in overlapping word ‘(50mg/L)’
Response 11: Thanks for your kind suggestion. We have given the space between number and unit.
adjusted p-value
- Check the reference section. It appears the repetition of numbering from ref. no 48 and check in the text accordingly.
Response 11: Thanks for your kind suggestion. The repetition of numbering reference happened in the revised manuscript with ‘Track Changes’. This repetition has disappeared in the revised manuscript without ‘Track Changes’ after the change was accepted.
